# Optimization of Processing Parameters of Aluminum Alloy Cylindrical Parts Based on Response Surface Method during Hydromechanical Deep Drawing

Yufeng Pan and Gaoshen Cai *

School of Mechanical Engineering, Zhejiang Sci-Tech University, Hangzhou 310018, China;
202120601036@mails.zstu.edu.cn
* Correspondence: caigaocan@zstu.edu.cn; Tel.: +86-0571-868-43343

**Abstract:** Aluminum alloy has been proposed as one of the next generation of lightweight body structure materials, which is widely used in the main components of the aerospace field. In order to realize efficient and accurate forming of aluminum alloy cylindrical parts, the response surface method combined with finite element simulation was used to optimize the key processing parameters during the hydromechanical deep drawing process. Three processing parameters of friction coefficient, pressure rate, and fillet radius of the die were selected as the optimization variables, and the maximum thinning rate of cylindrical parts was selected as the optimization evaluation index. The Box–Behnken design was selected to design the experiment scheme. A quadratic response model between the maximum thinning rate and the processing parameters was established by the response surface analysis software Design Expert for experimental design and data analysis. The optimal processing parameter combination was obtained through this model. The results show that the optimal conditions of maximum thinning rate can be met when the pressure rate is 11.6 MPa/s, the friction coefficient is 0.15, and the fillet radius of the die is 8 mm. Finally, the experimental verification was carried out by using the optimized combination of process parameters. It was found that the error between the experimental results and the predicted simulation results was within 5%, and the cylindrical parts which met the quality requirements were finally formed.

**Keywords:** evaluation and optimization; response surface methodology; hydromechanical deep drawing; formability



## 1. Introduction

In recent years, with the rapid development of the modern industry represented by aviation, aerospace, and automobile, lightweight materials have been widely used in these fields. It has become the trend of modern industrial development to achieve structural lightweight using lightweight materials [1–3]. As a lightweight structural material, aluminum alloy has the advantages of high toughness, high strength, excellent corrosion resistance, and low density [4–6]. Therefore, aluminum alloys are widely used in the aerospace and automotive industry [7]. However, aluminum alloys have poor forming ability at room temperature. When such components are processed by traditional processes, defects such as wrinkling and cracking often occur, and the forming accuracy is low, which restricts their further application [8].

Hydromechanical deep drawing technology, as an advanced precision-forming technology, uses the pressure of liquid as a force transfer medium instead of the die [9–11]. Due to the involvement of the liquid pressure medium, the sheet is pressed against the mold under the action of the liquid pressure to form the part [12–14]. Compared with traditional processing technology, the plasticity and ductility of formed parts are significantly increased, and the tendency to rupture is slowed down. It can be used for both light materials that are difficult to deform at room temperature and light alloy materials that

have poor plasticity at room temperature while ensuring the quality of the formed parts, which is an advanced forming technology [15–17].

During the hydromechanical deep drawing process, the final form quality of the part is affected by many processing parameters, such as loading path, friction coefficient, hydraulic pressure, etc. [18,19]. In order to obtain the target parts with better-forming quality, many scholars have investigated the hydroforming process of parts. Reddy et al. [20] used the finite element analysis method to study the influence of the blank holder force on the forming parts with the ultra-deep drawing (EDD) alloy steel sheet as the research object. It was found that increasing the blank holder force can effectively suppress the wrinkling of the parts, but the excessive blank holder force will cause the parts to break at the cylinder wall and the corner of the punch. Lang et al. [21] investigated the effect of pre-bulging on the hydroforming process of irregular box plates with unequal height, and the flat bottom was studied by numerical simulation and experiment. It is found that pre-bulging has an important influence on the forming results. Aiming at the problem of uneven thickness distribution in the hydroforming process of variable diameter tubes, Han et al. [22] investigated the influence of loading path on the shape of preformed parts by combining numerical simulation and experiment. It is found that the loading path plays an important role in the formation of beneficial folds. Only when the loading path is appropriate can beneficial wrinkles be formed. Cai et al. [23] carried out warm/hot sheet bulging tests of 2A16-O aluminum alloy by using elliptical bulging dies under various temperatures and pressure rates in an effort to investigate the macroscopic and microscopic influence of the pressure rate on the formability and microstructural evolution of hydro-bulging parts during warm/hot sheet hydroforming. It was found that the forming limit of the aluminum alloy was clearly influenced by the pressure rate as the temperature rose, wherein a lower pressure rate resulted in a higher-forming limit. However, in the actual forming process, the forming quality of the parts often involves the interaction between multiple parameters [24–26]. Empirical or trial-and-error methods of adjusting the parameters of the process can no longer meet the high-quality production of the parts. Using advanced optimization methods, useful rules can be excavated from a large amount of data, which is of great application value for guiding production [27,28].

In this study, aluminum alloy cylindrical parts were taken as the research object. Based on the hydromechanical deep drawing process, the efficient and accurate forming of cylindrical parts was investigated by using the finite element model combined with the optimization method–response surface method (RSM). The rate and fillet radius of the die were chosen for optimization. Based on the response surface method, the experimental data were analyzed. A quadratic response model between the maximum thinning rate of the cylindrical part and the processing parameters was established. The optimal processing parameter combination was obtained through this model. Finally, the reliability of the established response model and the optimized processing parameter combination was verified by the experiments.

## 2. Materials and Methods

### 2.1. Material Performance Parameters and Structural Characteristics of the Parts

The structural properties of the cylindrical part are shown in Figure 1. Among them, the top diameter $\Omega 1$ is 100 mm, the bottom diameter $\Omega 2$ is 134 mm, the bottom fillet radius is 5 mm, and the flange fillet radius is 8 mm. The blank material is an AA6013 aluminum alloy sheet with a diameter of 200 mm and a thickness of 1.0 mm. Based on the tensile data obtained from the uniaxial tensile test, the Swift hardening model is used to fit the true stress-true strain curve. The mechanical properties of the AA6013 aluminum alloy are shown in Table 1.

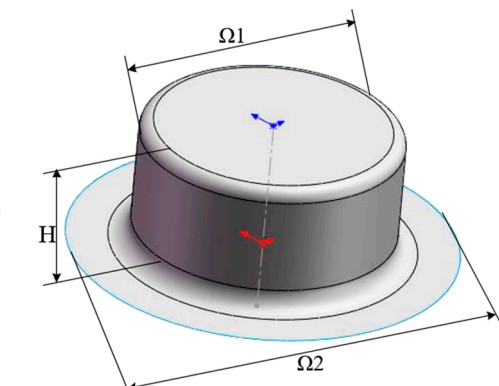

**Figure 1.** Geometric model of target forming parts (mm).

**Table 1.** Material parameters of AA6013 aluminum alloy.

| Material Name | E/MPa | Yield Limit $\sigma_s$/MPa | Tensile Strength $\sigma_b$/MPa | Poisson Ratio μ | Anisotropic Parameters | | | Hardening Index n | Hardening Factor K/MPa |
| --- | --- | --- | --- | --- | --- | --- | --- | --- | --- |
| | | | | | $r_0$ | $r_{45}$ | $r_{90}$ | | |
| AA6013 | 69,000 | 380 | 350 | 0.33 | 0.684 | 0.684 | 0.684 | 0.226 | 608.8 |

*2.2. Establishment and Simulation of the Finite Element Model*

The cylindrical part is formed based on the process of hydromechanical deep drawing, where the liquid is filled in the die as a liquid chamber. When the punch goes down, the liquid in the liquid chamber generates relative pressure so that the blank is tightly attached to the punch, and fluid lubrication can be generated between the die and the lower surface of the blank sheet. The main difference between hydromechanical deep drawing and classical drawing is the application of fluid pressure to the bottom surface of the workpiece. A three-dimensional stress field (liquid pressure provides thick stress) is generated due to the participation of a liquid medium during hydromechanical deep drawing, which improves the formability of the sheet. This sheet-forming process has several advantages, such as high dimensional accuracy, low tooling costs, proper uniformity, the ability to form complex parts, and better surface quality compared to classical drawing.

Firstly, the mold model is established in the three-dimensional software SolidWorks. The mold model is saved in an IGS format and imported into the sheet-metal forming professional software DYNAFORM. The forming process principle and finite element model are shown in Figure 2. The sheet thickness was set to 1 mm. The material was chosen to be the AA6013 aluminum alloy. After the sheet is set, the blank is meshed. The software Dynaform provides a variety of mesh division methods. There are two commonly used mesh division methods: one is plane sheet mesh division, and the other is surface mesh division. The plane sheet meshing is adopted, and its element type is the shell element. The shell element can realize double-sided contact and detection. During hydromechanical deep drawing, the sheet makes contact with the blank holder on one side of the flange, and the other side makes contact with the die. In the liquid chamber, one side of the sheet is in contact with the liquid, and the other side is in contact with the punch. Therefore, the sheet was divided by plane sheet mesh. The 4-node Belytschko–Tsay shell element is selected for the sheet metal. The punch, die, and blank holder are all regarded as rigid bodies, and the meshing was used by surface meshing. The rigid 4-node element is selected for the discretization of the die. The sheet was divided by plane sheet mesh. The punch, die, and blank holder are all regarded as rigid bodies, and the meshing was used by surface meshing. The rigid 4-node element is selected for the discretization of the punch, die, and blank holder. During the hydromechanical deep drawing process, the parameters were set as follows: a gap of 1.1 times the sheet thickness between the die, a gap of 1.2 times the sheet thickness between the punch and the binder, the blank size was a circular blank with a diameter of 200 mm, the friction coefficient between the punch and sheet was 0.3, the

friction coefficient between the die and the sheet was 0.075, the maximum liquid chamber pressure was 10 MPa, the value of the blank holder was 110 KN, and the drawing speed varied depending on the pressure rate.

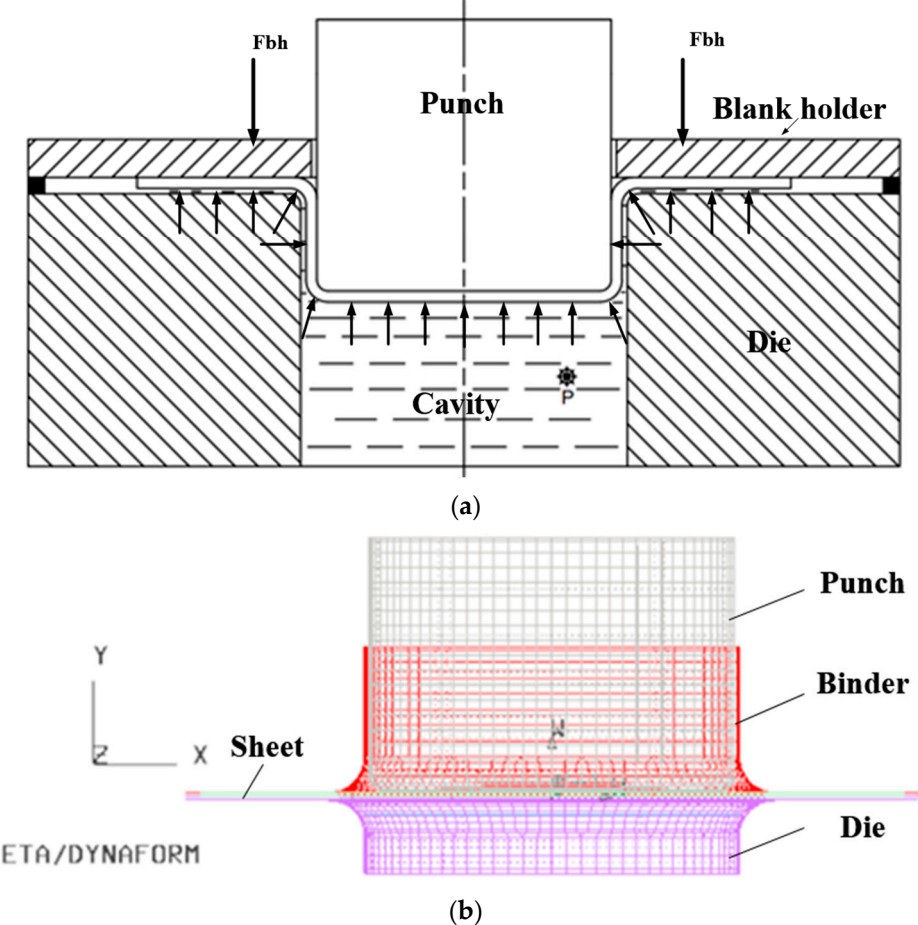

(a)

(b)

**Figure 2.** Forming principle and finite element model. (**a**) The principle of hydromechanical deep drawing. (**b**) Finite element model.

## 3. Simulation Results and Discussion

### 3.1. Effect of Friction Coefficient on Part Forming

During sheet hydroforming, the liquid presses the sheet on the surface of the punch, and the punch drives the sheet to draw down so as to improve the phenomenon that the simply pressed sheet does not stick to the die and promote the deep drawing. Therefore, the friction coefficient between the mold and the part plays a key role in controlling the flow of the metal sheet material and affecting the quality of the part forming. In actual production, lubricants are generally used. Hu et al. [29] investigated the approximate range of friction coefficients that can be achieved with different lubricants (including PhoS, MoS2, PTFE, deep drawing oil, etc.). Based on their findings, to study the influence of the friction coefficient between the blank holder and the sheet on the formability of parts and ensure that other processing parameters remain unchanged, the friction coefficients between the blank holder and the sheet are set to 0.09, 0.1, 0.2, and 0.3, respectively. The results of the axial stress distribution and the maximum thinning rate of the parts under different friction coefficients are shown in Figures 3 and 4.

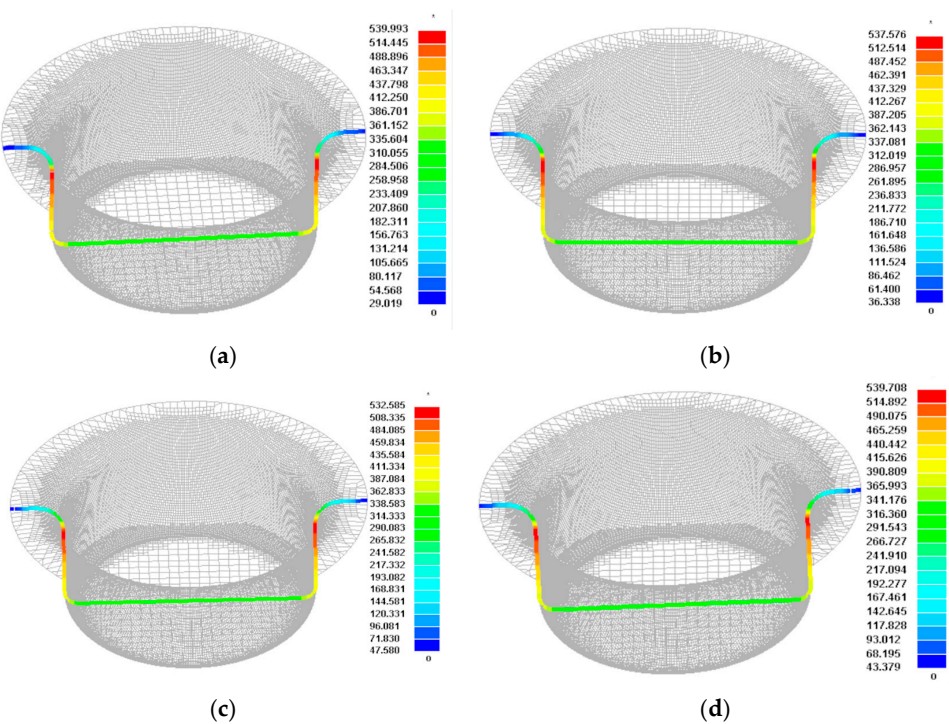

**Figure 3.** Axial stress distribution of forming parts under different friction coefficients. (**a**) 0.09, (**b**) 0.1, (**c**) 0.2, (**d**) 0.3.

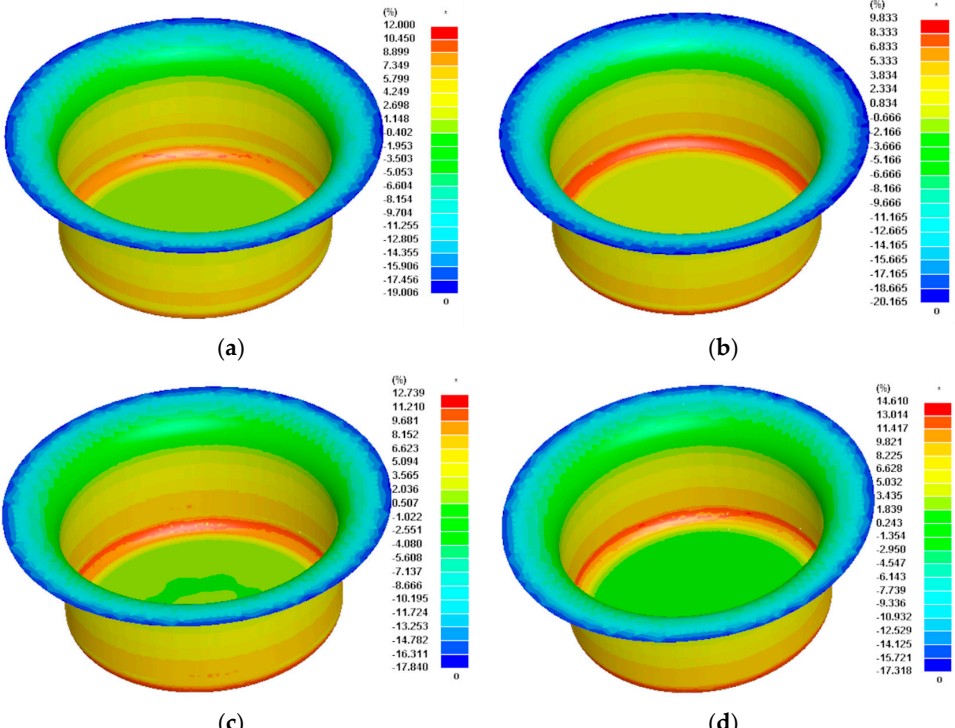

**Figure 4.** Maximum thinning rate of forming parts under different friction coefficients. (**a**) 0.09, (**b**) 0.1, (**c**) 0.2, (**d**) 0.3.

It can be seen that as the friction coefficient increases, the maximum axial stress of the part decreases, indicating that as the friction coefficient increases, the maximum axial stress of the cylinder wall can be reduced. As the friction coefficient further increases, the maximum axial stress of the cylinder wall begins to increase. With the increase of the

friction coefficient, the maximum thinning of the part decreases first and then increases, indicating that the increase of the friction coefficient is beneficial to the friction retention effect of the hydromechanical deep drawing. However, if the friction coefficient is too large, it will lead to a decrease in the surface finish of the mold and may cause damage to the surface of the sheet.

### 3.2. Effect of Pressure Rate on Part Forming

Pressure rate is one of the key parameters affecting the hydroforming performance of sheet metal. In the process of sheet hydroforming, the sheet metal is pressed into the liquid chamber by the punch during the drawing process, and the flow of the material makes the performance of the formed part change. The change rate of fluid pressure in the mold cavity will affect the material flow rate. Strain rate is the change of material strain (deformation) relative to time. Strain rate is usually defined as the derivative of strain relative to time. Strain rate is a measure of material deformation rate. Pressure rate is the increment of liquid pressure per unit time. In other words, in the hydroforming process, plastic deformation with different pressure increments will obtain different strain increments [30]. That is to say, the pressure rate is used to characterize the speed of material deformation, which in turn affects the formability of the formed parts. Therefore, in the process of hydromechanical deep drawing, the pressure rate will affect the forming performance of the formed parts. In order to study the influence of pressure rate on the forming performance of the target part and ensure that other process parameters are constant, the pressure rate is 1.25–4.75 MPa/s for analysis. The results of the axial stress distribution and the maximum thinning rate of the parts under different friction coefficients are shown in Figures 5 and 6.

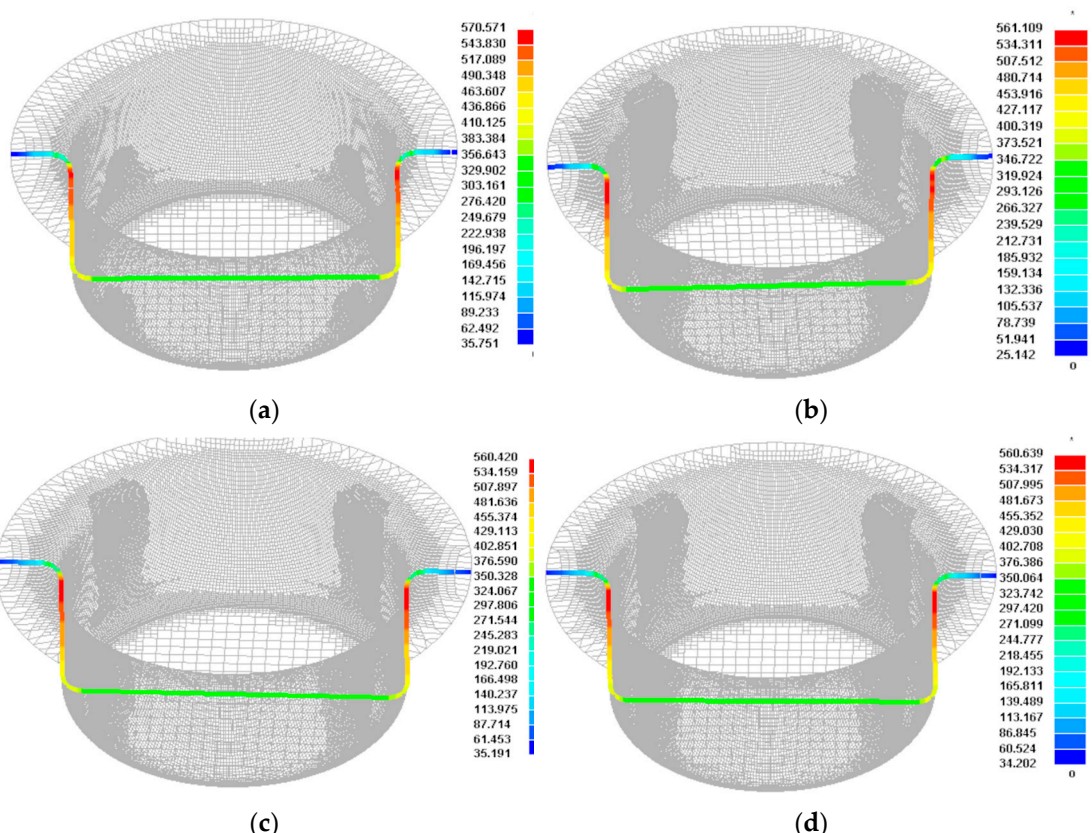

**Figure 5.** Axial stress distribution of forming parts under different pressure rates. (**a**) 1.25 MPa/s, (**b**) 2.5 MPa/s, (**c**) 3.75 MPa/s, (**d**) 4.75 MPa/s.



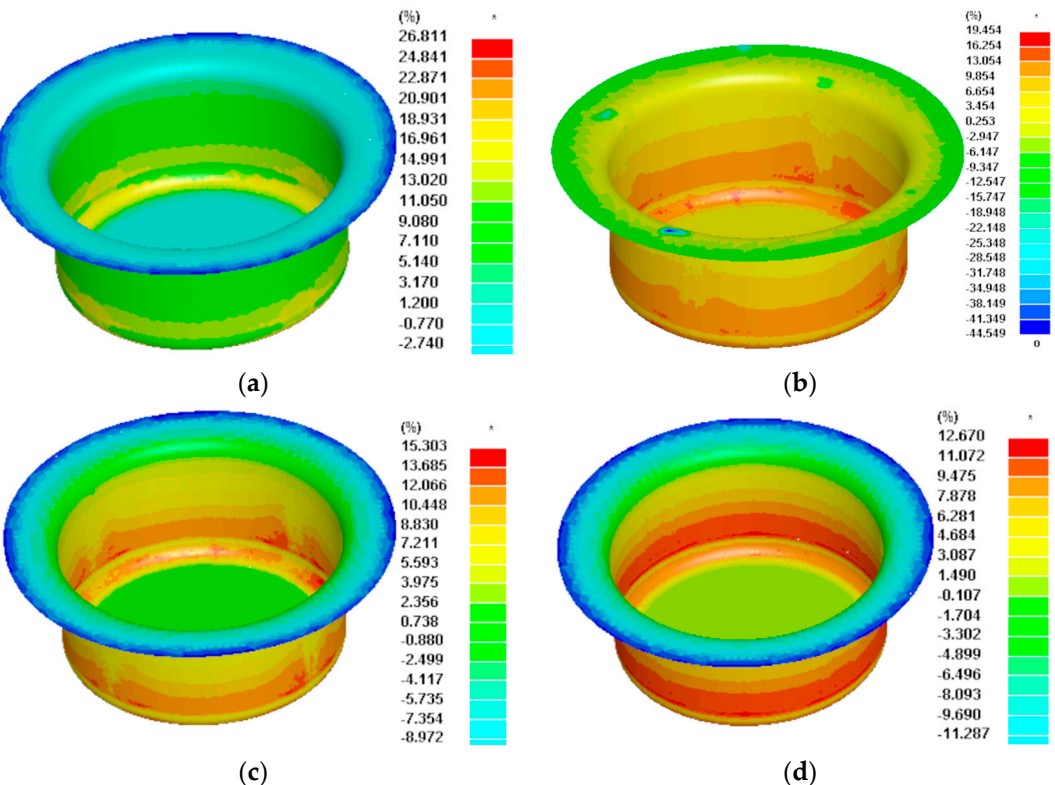

**Figure 6.** Maximum thinning rate of forming parts under different pressure rates. (**a**) 1.25 MPa/s, (**b**) 2.5 MPa/s, (**c**) 3.75 MPa/s, (**d**) 4.75 MPa/s.

From the diagram, it can be seen that with the increase of the pressure rate, the maximum axial stress and the maximum thinning rate of the parts are becoming smaller and smaller, indicating that the forming effect of the aluminum alloy cylindrical parts will be improved under the larger pressure rate. The overall deformation is relatively uniform, and the forming effect is better.

*3.3. Effect of Fillet Radius on Part Forming*

The fillet radius of the die is an important processing parameter in the hydromechanical deep drawing process, which affects the forming performance of the parts. Similarly, to ensure that other forming parameters remain unchanged, the forming parts under different die fillets radius are analyzed, respectively. The maximum thinning rate of the forming parts under different fillets radius is shown in Figure 7. It can be seen from the figure that with the increase of the die fillet radius, the maximum thinning rate of the formed parts decreases first and then increases. When the fillet radius is too small, the material has a large bending deformation. The bending resistance and friction force increase, the drawing force increases, and the parts become thin and serious, resulting in cracking. With the increase of the die fillet radius, the maximum thinning rate of the formed part also decreases, and the smaller the tensile stress of the sheet at the die fillet, the more conducive to deep drawing. When the die fillet radius is 10 mm, the maximum thinning rate of the part is the smallest. When the fillet radius is greater than 10 mm, the maximum thinning rate of the parts increases with the increase of the fillet radius. This is because, with the increase in the fillet radius, the area where the blank is not stressed becomes larger, which is prone to wrinkling.

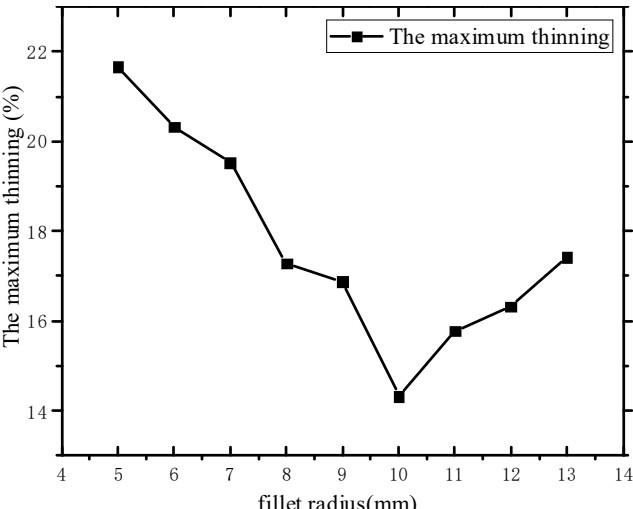

**Figure 7.** Variation trend of maximum thinning rate under different fillet radii.

## 4. Experimental Results and Discussion

### 4.1. Experimental Factors and Design

As can be seen from the above analytical results, the maximum thinning rate of the part is greatly affected by the friction coefficient, pressure rate, and fillet radius of the die. Moreover, the forming quality of parts often involves the interaction between multiple parameters. The process parameters can be optimized by the response surface method to obtain the process parameters that can form the optimal forming quality of the target parts and provide full play to the potential of the process. In order to better describe the forming performance of parts, the three factors of friction coefficient (A), pressure rate (B), and fillet radius (C) in the process of hydroforming are taken as independent variables, and the maximum thinning rate is taken as the response value. The response surface experiment was designed according to the principle of the Box–Behnken design (BBD) central composite experiment and the experiment was designed by Design Expert. The design of the experimental factors and levels are shown in Table 2. Through numerical simulation analysis and processing parameter optimization, the optimum process conditions were determined.

**Table 2.** Experimental factors and levels.

| Level | Factor | | |
|:---:|:---:|:---:|:---:|
| | **A Friction Coefficient** | **B Pressure Rate/ MPa/s** | **C Fillet Radius/mm** |
| −1 | 0.09 | 1.25 | 5 |
| 0 | 0.295 | 6.75 | 9 |
| 1 | 0.5 | 12.5 | 13 |

### 4.2. Experimental Results of the Box–Behnken Design

The maximum thinning rate values were measured for different formation parameters depending on the software design conditions. The experimental design and results of the response surface are given in Table 3. Taking the maximum thinning rate value as the index, the experimental results in Table 4 are analyzed using Design-Expert software, and the regression equation of the maximum thinning rate is obtained by a quadratic polynomial fitting, as shown in Formula (1):

$$\begin{aligned}
The\ maximum\ thinning\ rate = {}& 18.04 + A - 0.575 \times B + \\
& 0.375 \times C - 0.45 \times A \times B - 0.05 \times A \times C + 0.1 \times B \times C \\
& + 0.68 \times A^2 - 1.12 \times B^2 + 1.18 \times C^2
\end{aligned} \tag{1}$$

**Table 3.** Response surface experimental design and results.

| Test Number | A Friction Coefficient | B Pressure Rate (MPa/s) | C Fillet Radius (mm) | The Maximum Thinning (%) |
|---|---|---|---|---|
| 1 | 0.09 | 1.25 | 9 | 16.7 |
| 2 | 0.5 | 1.25 | 9 | 19.8 |
| 3 | 0.09 | 12.25 | 9 | 16.3 |
| 4 | 0.5 | 12.25 | 9 | 17.6 |
| 5 | 0.09 | 6.75 | 5 | 18.3 |
| 6 | 0.5 | 6.75 | 5 | 20.2 |
| 7 | 0.09 | 6.75 | 13 | 19.7 |
| 8 | 0.5 | 6.75 | 13 | 21.4 |
| 9 | 0.295 | 1.25 | 5 | 18.6 |
| 10 | 0.295 | 12.25 | 5 | 17.4 |
| 11 | 0.295 | 1.25 | 13 | 18.6 |
| 12 | 0.295 | 12.25 | 13 | 17.4 |
| 13 | 0.295 | 6.75 | 9 | 18.6 |
| 14 | 0.295 | 6.75 | 9 | 17.8 |
| 15 | 0.295 | 6.75 | 9 | 18.4 |
| 16 | 0.295 | 6.75 | 9 | 17.3 |
| 17 | 0.295 | 6.75 | 9 | 17.6 |

**Table 4.** Quadratic model variance analysis results of minimum thickness.

| Source | Sum of Squares | df | Mean Square | F-Value | *p*-Value |
|---|---|---|---|---|---|
| Model | 25.20 | 9 | 2.80 | 6.40 | 0.00115 (significant) |
| A-friction | 8.00 | 1 | 8.00 | 18.29 | 0.0037 |
| B-pressure rate | 2.65 | 1 | 2.65 | | 0.00435 |
| C-fillet radius | 1.13 | 1 | 1.13 | 2.57 | 0.00152 |
| AB | 0.8100 | 1 | 0.8100 | 1.85 | 0.2158 |
| AC | 0.0100 | 1 | 0.0100 | 0.0229 | 0.8841 |
| BC | 0.0400 | 1 | 0.0400 | 0.0914 | 0.7711 |
| A2 | 1.95 | 1 | 1.95 | 4.45 | 0.0728 |
| B2 | 5.28 | 1 | 5.28 | 12.07 | 0.0103 |
| C2 | 5.86 | 1 | 5.86 | 13.40 | 0.0081 |
| Residual | 3.06 | 7 | 0.4374 | | |
| Lack of fit | 0.7300 | 3 | 0.2433 | 0.7506 | not significant |
| Pure Error | 2.33 | 4 | 0.5830 | | |
| Cor Total | 28.26 | 16 | | | |
| R2 = 0.9427 | R2adjust = 0.9345 | | | | |

### 4.3. Analysis of Variance

The hydroforming process is a typical material composite forming process. The variance regression analysis (ANOVA) of the maximum thinning rate is shown in Table 4. When other parameters remain unchanged, the maximum thinning rate distribution of the part is mainly determined by the blank holder force, friction coefficient, and pressure rate. As can be seen from Table 4, when the *p* = 0.00115 of this model, the response surface model was significant. The coefficient of variation was less than 10%, indicating that non-experimental factors had little effect on the results, and the model had good experimental stability. The correlation coefficient R2 of the model is 0.9345, and the corrected correlation coefficient R2adj is 0.9427, indicating that the experimental accuracy is high. Therefore, the model can be used to analyze and predict the influence of different forming parameters on the minimum wall thickness in hydroforming.

The normal probability distribution of residuals is shown in Figure 8. It can be seen that each residual order point is roughly distributed around a straight line, indicating that the model fits well and the experimental design is reliable. The relationship between the

predicted value and the actual value of the minimum wall thickness is shown in Figure 9. It can be seen that the points of the predicted value and the actual value are roughly distributed in the same straight line, indicating that the predicted value of the response model is close to the actual value, and the error is basically negligible. The quadratic response model of the minimum wall thickness can accurately predict the actual value in the forming process.

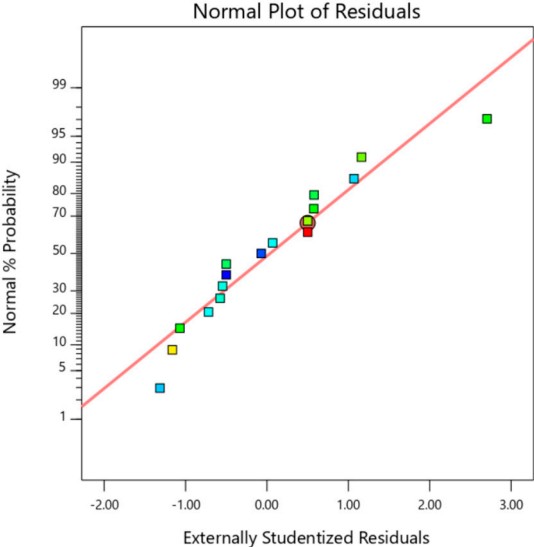

**Figure 8.** Normal probability of model residuals.

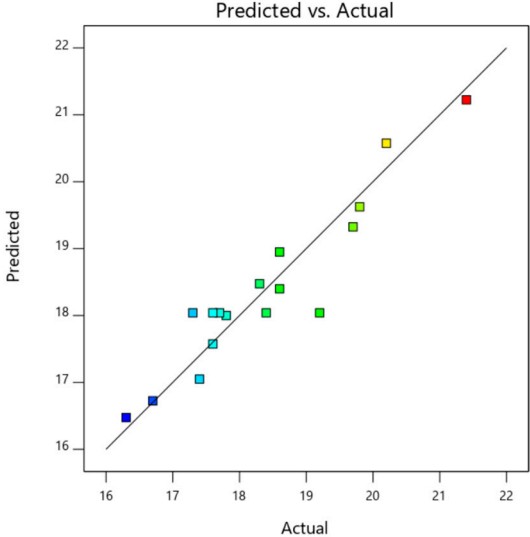

**Figure 9.** Distribution of residuals and predicted values.

### 4.4. Response Surface Results and Interaction Effects Analysis

The response surface is shown in Figure 10, which can intuitively analyze the interaction law of each process parameter on the response amount. It can be clearly seen from the figure that, compared with the response surface of AC and BC, the response surface of AB is relatively flat. It shows that the interaction between pressure rate and friction coefficient has little effect on the maximum thinning rate.

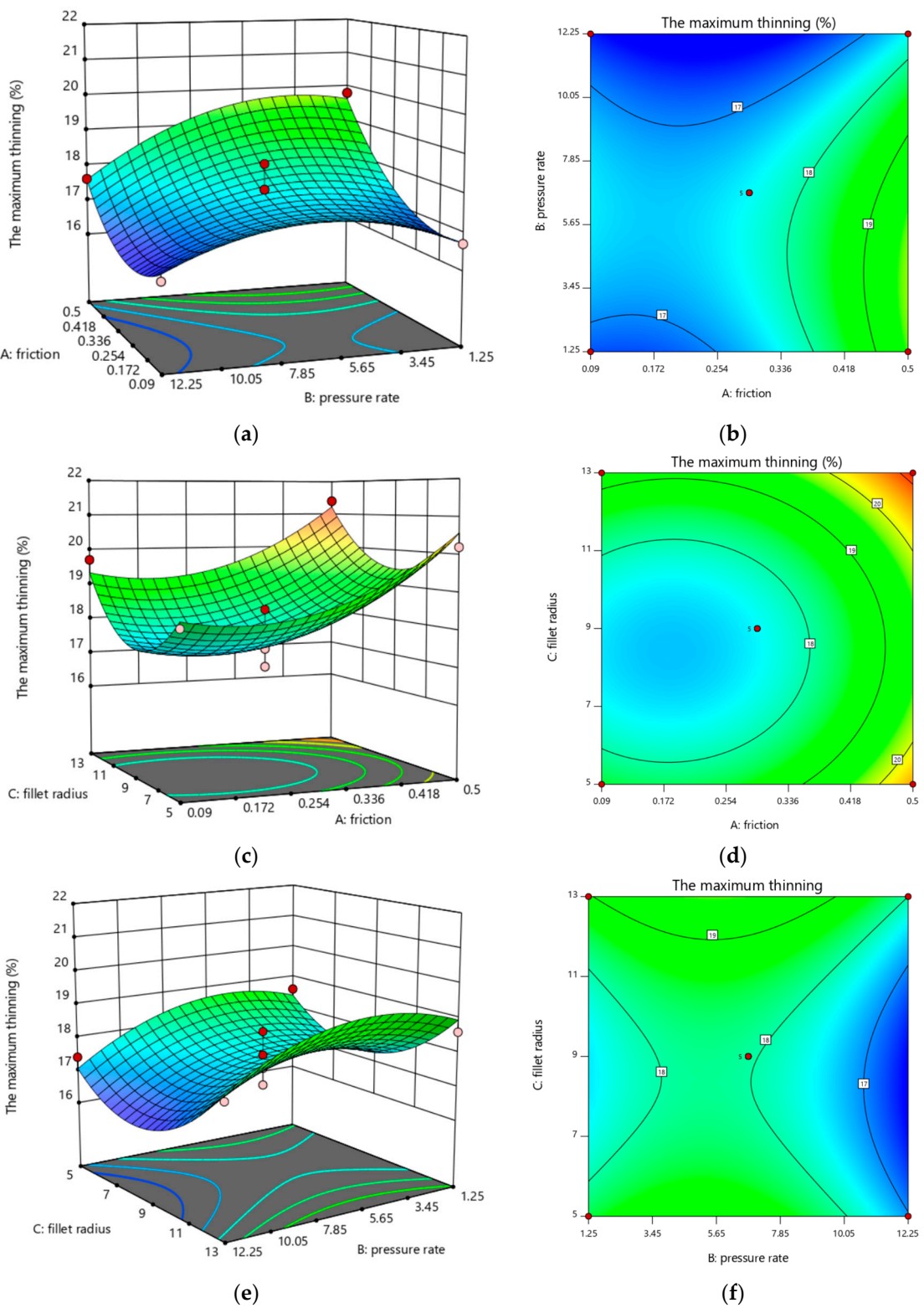

**Figure 10.** Response surface and contour maps of formability interaction: (**a**,**b**) response surface and contour maps of friction (A) and pressure rate (B); (**c**,**d**) response surface and contour maps of friction (A) and fillet radius (C); (**e**,**f**) response surface and contour maps of pressure rate and fillet radius. (**a**) (A, B), (**b**) (A, B), (**c**) (A, C), (**d**) (A, C); (**e**) (B, C), (**f**) (B, C).

### 4.5. Experimental Verification and Optimization

Through analysis and verification, the best process combination is obtained. The optimal processing parameters of the quadratic response model between the maximum thinning rate of the part and the processing parameters were obtained using the software Design Expert: the pressure rate was 11.6 MPa/s, the friction coefficient was 0.15, and the fillet radius was 8 mm. The experiment was carried out on the YRJ-50t bulging–drawing machine developed by the Beijing University of Aeronautics and Astronautics [31], as shown in Figure 11. The wall thickness of the formed part is measured by the ultrasonic thickness gauge. As shown in Figure 12, the accuracy error between the simulation results and the experimental results is within 5%, which has high accuracy. The parts formed by optimized simulation parameters are shown in Figure 13. The response surface method is used to optimize the forming parameters for hydroforming. This method can also be applied to other typical parts.

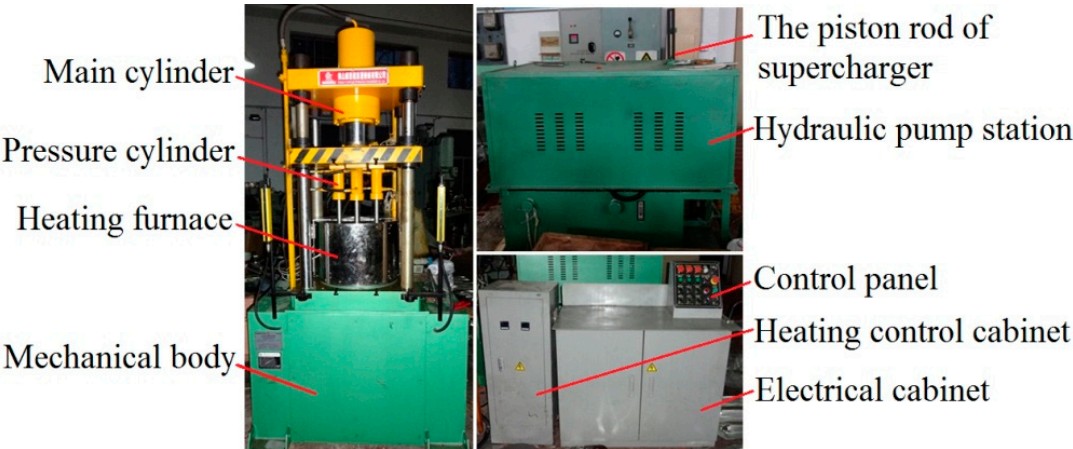

**Figure 11.** Hydraulic bulging–drawing machine [31].

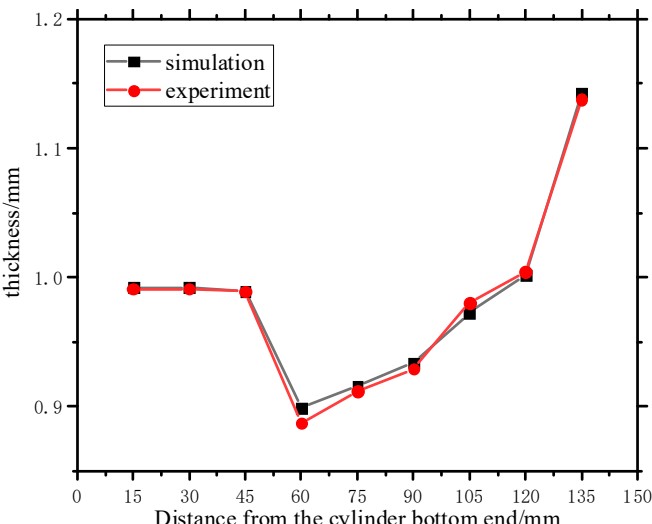

**Figure 12.** Thickness distribution for experimental and simulation comparison.

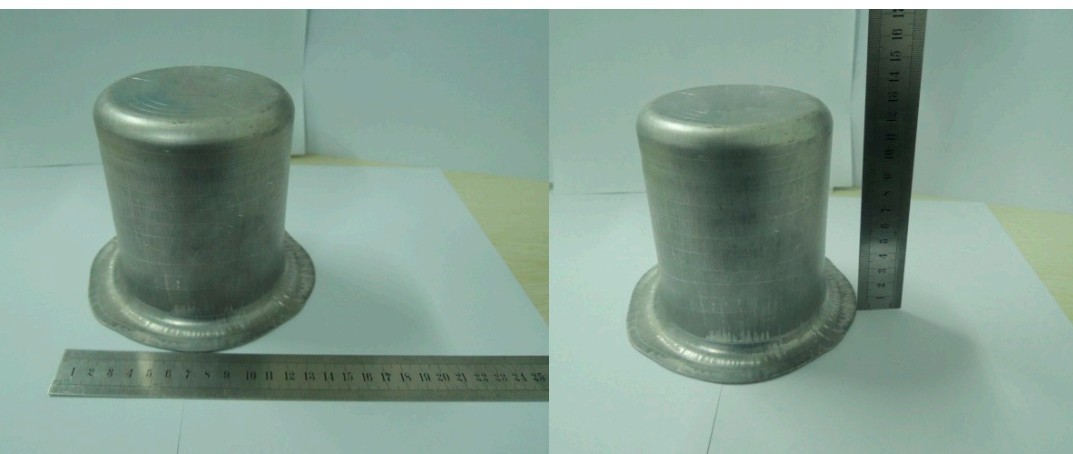

**Figure 13.** Formed part.

## 5. Conclusions

In this study, aluminum alloy cylindrical parts were taken as the research object. Based on the hydroforming deep drawing process, the maximum thinning rate was taken as the optimization evaluation index, and the three process parameters of friction coefficient, pressure rate, and fillet radius were selected for optimization. The finite element model combined with the optimization method–response surface method (RSM) was used to conduct experiments according to the Box–Behnken design. Then, the effects of the considered parameters on the formability were discussed based on RSM and ANOVA. The main conclusions are as follows:

(1) The friction coefficient, pressure rate, and fillet radius of the die have a significant influence on the maximum thinning rate and axial stress distribution of the parts. With the increase of the friction coefficient and fillet radius of the die, the maximum thinning rate of the parts decreases first and then increases. With the increase in the pressure rate, the maximum axial stress and the maximum thinning rate of the parts decrease and finally remain unchanged;

(2) The quadratic response model between the maximum thinning rate and processing parameters was established. Through the quadratic response model, the optimal combination of processing parameters was obtained: the pressure rate was 11.6 MPa/s, the friction coefficient between the blank holder and the sheet was 0.15, and the fillet radius of the die was 8 mm;

(3) The experimental results show that the quadratic regression model can satisfactorily estimate the correlation between formability and the considered process parameters, and the error range is less than $\pm 5\%$.

**Author Contributions:** Conceptualization, Y.P. and G.C.; funding, G.C.; investigation, Y.P. and G.C.; writing—original draft preparation, Y.P.; writing—review and editing, Y.P. and G.C.; methodology, Y.P. and G.C.; supervision, G.C.; visualization, Y.P. and G.C. All authors have read and agreed to the published version of the manuscript.

**Funding:** This work was supported by the Fundamental Research Funds of Zhejiang Sci-Tech University, Grant No. 22242297-Y, the Zhejiang Provincial Natural Science Foundation of China, Grant No. LQ18E050010, and the China Postdoctoral Science Foundation Funded Project, Grant No. 2018M642482.

**Data Availability Statement:** Data will be made available upon request.

**Conflicts of Interest:** The authors declare that they have no known competing financial interest or personal relationships that could have appeared to influence the work reported in this paper.

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
