# Peer review of "Optimization of Processing Parameters of Aluminum Alloy Cylindrical Parts Based on Response Surface Method during Hydromechanical Deep Drawing"

_metals, doi:10.3390/met13081406_

Round 1

Reviewer 1 Report

In the manuscript entitled “Optimization of processing parameters of aluminum alloy cylindrical parts based on response surface method during hydromechanical deep drawing” the response surface method and finite element simulation were combined to investigate the efficient and accurate forming of aluminum alloy cylindrical parts. The manuscript is well written and discussed but needs adjustments before publication. See the recommendations below:

·         It is necessary to insert a section entitled: Materials and Methods. Some techniques, materials, and software are described in the wrong place.

·         The numbers of all figures need to be updated.

·         The authors should increase the resolution of the legend in Fig. 6.

·         Also, in some figures, the letters of legends are too small.

Reviewer 2 Report

The topic of the article is current. With hydromechanical deep drawing process, it is possible to improve the formability of materials and the quality of drawn parts compared to classical deep drawing process. There are several simplifying formulations in the introductory part.

In line 41, the authors state: "Compared with traditional processing technology, the forming limit, plasticity and ductility of the drawn parts will be greatly increased, the yield strength will be reduced, and the tendency to rupture will be slowed down".

It is necessary for the authors to explain what they mean by the term "forming limit". Is it the forming limit diagram FLD or the limiting drawing ratio- LDR? It is also necessary to explain why the yield strength is reduced during hydromechanical drawing? It is a property of the material that is determined by the structure of the material. Is it possible to reduce the yield stress by hydromechanical pulling? If so, an explanation must be added. How?

 In tab. 2, it is necessary to supplement the description of what ss  and sb are.

In tables 2, 3, 4, it is necessary to complete the individual parameters of the unit and insert them from the parenthesis.

In hydromechanical deep drawing, frictional forces have different meanings. Some positively affect the hydromechanical deep drawing process and others negatively affect the drawing process. Add to figure 3 individual components of the total traction force during hydromechanical pulling of cylindrical ones and state the equation ( FTHD=Fid+..). Add, what is the difference in the force effect of hydromechanical dep drawing compared to classic deep drawing.

In the picture, the term Binder is not used correctly - it is a blank holder for a sheet blank.

 The thinning of the drawn part wall thickness and formability depend on the stress and deformation scheme in individual parts of the yield. Therefore, it is necessary to supplement, for example, on figure 2 and describe how the stressn and deformation scheme in hydromechanical deep drawing differs from the scheme in classical drawing.

The scheme of the tool and the description of its function are very simply described. The average reader can hardly imagine how the tool works and what the advantages of hydromechanical deep drawing are.

 In lines 85 to 96, in chapter 2.2 Determination and simulation of finite elements - it is necessary to add which input parameters were entered into software DYNAFORM - for example, indicate prt sc or  them describe. For example, are the material data listed in Table 2 enough for us?

During hydromechanical deep drawing, the sheet metal does not come into contact with the drawing edge of the die. Add description why friction coefficients 0.09, 0.1, 0.2, 0.3 were used. Describe whether different values ​​of the friction coefficients were entered for different contact surfaces (between the blank holder and the sheet, between the sheet and the drawing edge of the die, etc.) or only one for all contact surfaces. As I mentioned above, friction on individual contact surfaces has different effects on the process of hydromechanical deep drawing process.

 In lines 128-131, the authors state that: „The strain rate  can be characterized by pressure rate (that is, the deformation  speed of the material during the forming process can be represented by pressure rate),  which in turn affects the forming performance of the formed part.“

It is necessary to explain what "The strain rate" is. It is most appropriate to state the relationship for the determination of "The strain rate" .

Similarly, it is necessary to explain the term "Pressure rate", since a precise sheme of the hydromechanical deep drawing process is not given, it is difficult to find relationships between strain rate and pressure rate.

  Figure 8 - it is necessary to add at the pressure of the liquid the results were obtained.

Figure 13 – on the x-axis.

 In line 168-172, the authors state that:  „ When the die fillet radius is 10 mm, the maximum  thinning rate of the part is the smallest. When the fillet radius is greater than 10 mm, the  maximum thinning rate of the part increases with the increase of the fillet radius. This is  because the increase of the fillet radius, the area where the blank is not stressed becomes  larger, which is prone to wrinkling.“

Wrinkling on the drawn parts in classical drawing occurs when the radius of die of the die is greater than 10 mm, which means that a significant part of the sheet is not supported, and even with increased pressure of the blank holder, it is not possible to effectively remove the wrinkling on the drawn parts. One of the advantages of hydromechanical drawing is that by the pressure of the liquid on the unsupported surface of the sheet can effectively eliminate wrinkling on the drawn parts even with radii greater than 10 mm.

 In chapter 4.1 Experimental factors and design, the authors describe the DoE experiment. The experiment was appropriately planned and the results were adequately tested - prediction relationship (1).

 In lines 246 and 247 the authors mention the term "Design Expert" is this correct? Or it is more appropriate to use the term Design of Experiments (DoE)  In lines 244-248, the authors state that: “Through analysis and verification, the best combination of processes is obtained. The optimal processing parameters of the quadratic response model between the maximum thinning rate of the part and the processing parameters were obtained using Design Expert: the pressure rate was 11.6 MPa/s, the friction coefficient was 0.15, and the fillet radius was 8 mm. " Is it necessary to explain (add) why the optimal coefficient of friction for hydromechanical deep drawing is 0.15? Friction in the liquid is very small. How is it possible to ensure a coefficient of friction of 0.15 during hydromechanical drawing? In Figure 13, it is necessary to add units on the horizontal axis and to indicate the units in a round bracket. 

Reviewer 3 Report

Notes:

1- In abstract, it is preferable to mention the thickness of the sheet in the deep drawing process.

2- No reference was made in the abstract to the type of aluminum alloy AA6013.

3- The last section of Paragraph (1. introduction) that includes the aim of the paper is very long (14 lines), it contains details that do not need to be mentioned in this paragraph, it is shortened by focusing on the main goal without details.

4- In Paragraph 2.1, there is a clear error in mentioning the diameters between figure (1) and text of paragraph  i.e. 100 and 134 mm, where, in the text mentioned reversed.

5- Table No. (1),  chemical composition of AA6013 aluminium alloy , it was preferred measured by the author(s) and compares it with a standard.

6- The author(s)  did not clarify whether the mechanical properties of the aluminum alloy shown in Table No. (2) were by the author(s)  or from a reference , if from a reference must be mentioned this reference .

7- Figure No. 3 (a) It is preferable to indicate in the text or chart the value of the gap between punch and die, and how it was calculated.

8- It is preferable refer to the inputs and values ​​that have been entered into the simulation program.

9- What are the initial conditions, boundary conditions, and assumptions of return requests that were adopted in the numerical simulation?

10- What is the type of the elements that was used in meshing , and why it’s selected.

11- What is the value of pressure applied on the blank holder?

-

Round 2

Reviewer 1 Report

All inquiries were answered; accepted for publication.

Reviewer 2 Report

Dear Authors,

the specification of goalis acceptable.

You have verbally described the stress-deformation state - it is possible to accept it.

In diffrent drawn parts are diffrent stress-strain schemes, so it is necessary to draw them in the picture.

In the modified version, I did not find a description of the individual components of the forces that made up the total force, which, in my opinion, is crucial for assessing the impact on formability.

On the figure 1 also does not show the individual force components. What is the value of friction on the edge die where there is no contact between the sheet and the die?

I see the biggest problem in the result, where you indicate the value of the coefficient of friction at the end of 0.15. I wonder what is the point of hydromechanical deep drawing? 

The purpose of optimization is to achieve maximum formability - minimum thinning.

Does thinning depend only on friction, pressure and tool radius?

The properties of the material do not affect the thinning of the  wall of drawn part.

Table 1 is missing units for K.
